# Reviews and syntheses: The role of process-based modeling of the $CO_2$:$CH_4$ production ratio in predicting future terrestrial Arctic methane emissions

Marius Moser[1], Lara Kaiser[1], Victor Brovkin[2,3], and Christian Beer[1,3]

[1]Department of Earth System Sciences, Universität Hamburg, 20146, Hamburg, Germany
[2]Max Planck Institute for Meteorology, 20146, Hamburg, Germany
[3]Center for Earth System Research and Sustainability, Universität Hamburg, 20146, Hamburg, Germany

**Correspondence:** Marius Moser (marius.moser@uni-hamburg.de)

**Abstract.** Thawing permafrost in the Arctic threatens to potentially release large amounts of decomposed organic matter as $CO_2$ or $CH_4$ to the atmosphere. Predicting the ratio of emitted $CO_2$ to $CH_4$ is imperative for reliable future projections. Here, we review the recent literature concerning methanogenesis, and its current representation in both land surface models (LSMs) and the state-of-the-art process-based methane models. We found that the key processes, required to capture the dynamics of the $CO_2$:$CH_4$ production ratio, are: fermentation, hydrogenotrophic methanogenesis, and acetoclastic methanogenesis. Commonly discussed linked processes are Fe(III)-reduction and homoacetogenesis. Environmental factors influencing these processes, as identified in the literature, are: temperature, pH, water table position and alternative electron acceptors. While modern process-based methane models account for most of these factors and processes, the same is not true for the simplified methane formulations in many LSMs, which often opt for pre-set parameters that define a constant share of methane production from anaerobic decomposition. This static approach stands in opposition to the growing amount of lab and in-situ data, which suggest a high degree of spatio-temporal variability concerning this ratio, thus preventing its accurate prediction in a changing future Arctic. The challenge lies in upscaling the data as the environmental factors are barely quantified at the pan-Arctic scale. Additionally, there remains the important challenge of how to model and parameterize the temperature dependence of the individual underlying processes. Going forward, these challenges need to be overcome in order to reliably project the $CO_2$:$CH_4$ production ratio and methane emissions on larger scales. This will require a more process-based approach of methanogenesis in LSMs, for which we suggest a baseline concept here.

## 1 Introduction

Permafrost-affected soils are a significant global carbon pool, storing more carbon than there currently is in the atmosphere (Hugelius et al., 2014; Mishra et al., 2021; Friedlingstein et al., 2022). This permafrost is already beginning to thaw (Biskaborn et al., 2019) and large-scale future losses are projected (McGuire et al., 2018) due to climate change and the increased warming that is expected to occur in the Arctic (IPCC, 2021). Thawing permafrost enables the microbial decomposition of the large

amounts of carbon stored across the Arctic, potentially releasing considerable amounts of carbon to the atmosphere, thus creating a self-reinforcing carbon-climate feedback (Beer, 2008; Schuur et al., 2015, 2022).

Of particular interest is the form in which the carbon will be released to the atmosphere, namely as either $CO_2$ or $CH_4$, due to the strong difference in climate forcing between the two gases, with methane being the much more potent one (Myhre et al., 1998). Methane has contributed 11% to the total radiative forcing since 1960, despite its relatively low concentration in the atmosphere (Canadell et al., 2021). Furthermore, methane emissions have increased nearly 2-fold in the last two centuries (Canadell et al., 2021) and continue to grow persistently (Saunois et al., 2020), thus garnering much research interest (IPCC, 2021; Canadell et al., 2021; Saunois et al., 2020; Xu et al., 2016; Chandel et al., 2023). The majority of emissions are expected to occur as $CO_2$ (Miner et al., 2022; Schädel et al., 2016) but recent studies also highlight the importance of $CH_4$ emissions from a thawing Arctic (Knoblauch et al., 2018; Kleinen et al., 2021; Turetsky et al., 2020). This stresses the need for a more accurately constrained future methane budget, which presently remains uncertain (Ito et al., 2023). Methane production is tied to anoxic conditions in the soil, which usually occur when the soil becomes waterlogged (van Huissteden, 2021). Since the future hydrology of the Arctic remains uncertain (Andresen et al., 2020; de Vrese et al., 2023), so does the extent and timing of Arctic methane emissions (Canadell et al., 2021). This is also the reason for the relative scarcity of model studies on the topic that involve Earth System Models (ESMs) (de Vrese et al., 2021). In fact, many ESMs do not explicitly model $CH_4$ emissions at all (Schuur et al., 2022). Those who do, often represent methane production in a highly simplified way, frequently via a certain $CO_2$:$CH_4$ production ratio factor (Kleinen et al., 2020; Gasser et al., 2018; Riley et al., 2011). This is despite the fact that this ratio has been shown to be highly variable in both laboratory (Knoblauch et al., 2018; Heslop et al., 2019) and in situ studies (Galera et al., 2023). Knoblauch et al. (2018) showed in their long-term incubation study, that methanogenic communities in permafrost soils need time to establish themselves, resulting in a lag time of multiple years before eventually a $CO_2$:$CH_4$ ratio of 0.92±0.18 was reached (Knoblauch et al., 2018). Heslop et al. (2019) reported C-$CO_2$:C-$CH_4$ production ratios between 13-134, depending on soil depth, from their incubations. Galera et al. (2023) estimated in situ median $CO_2$:$CH_4$ emission ratios of 12 and 373, depending on the tundra type of polygonal tundra soils, though their values were affected by methanotrophy and, therefore, the actual production ratios are likely smaller (Galera et al., 2023). Another important factor, besides hydrology and soil properties, is vegetation. Due to the $CO_2$ fertilization effect, the plant productivity will potentially increase, providing additional substrate for the methanogens (Kettunen, 2003). This aspect will especially be important in the Arctic, where large-scale vegetation changes can be expected upon warming (Swann et al., 2010; Chapin et al., 2005; Cho et al., 2018).

The methane emission calculation does not stop at the methane production, however. For the methane to reach the atmosphere it needs to be first transported from its production point, through the soil column, to the surface. On its way to the surface, the methane can be oxidized by methanotrophic microbes in oxic soil layers, which affects the $CO_2$:$CH_4$ ratio at the surface (Wania et al., 2010). There exist three important transport mechanisms: diffusion, ebullition, and plant-mediated transport (Walter and Heimann, 2000; Wania et al., 2010; Kaiser et al., 2017). Their relative share is important with regards to the potential methane oxidation, since plant-mediated transport, e.g., can enable methane to bypass the oxidative soil layers (Knoblauch et al., 2015). Diffusion describes the methane transport along a concentration gradient and is the slowest way of transport,

thus facilitating methane oxidation (Knoblauch et al., 2015). Ebullition is a rather fast process, describing the rise of methane gas bubbles through water (Knoblauch et al., 2015). Lastly, plant-mediated transport happens largely through vascular plants, which possess so called aerenchyma, a type of aerated tissue responsible for supplying $O_2$ to the roots (Wania et al., 2010; Knoblauch et al., 2015). That tissue enables methane and $CO_2$ to be transported through the plant to the atmosphere (Wania et al., 2010). This connection between methane release and plants further hints at the fact that, aside from hydrological changes, future methane emissions are also influenced by vegetation changes (Kettunen, 2003). Many models account for these three transport ways, including large-scale land surface models (Kaiser et al., 2017; Wania et al., 2010; Riley et al., 2011; Chinta et al., 2024). In fact, Xu et al. (2016) found that the majority of methane models in their meta-study represented these three pathways already, albeit to varying degrees of complexity. Considering all this, it is worth looking into the recent developments concerning methane modeling. In this study we will focus on the methanogenesis aspect in particular, since other methane-related processes, e.g., methane transport, have already been implemented into models in more detail over the years (Wania et al., 2010; Kaiser et al., 2017; Xu et al., 2016). It is also worth noting that we focus on terrestrial emissions in this study. Other methane sources, e.g., wildfires, lakes, and marine and geological sources, also make up a significant part of the Arctic methane budget, potentially contributing over 30% to it (Parmentier et al., 2024). We will first recap the crucial processes and environmental factors that have been identified to govern methanogenesis and the $CO_2$:$CH_4$ ratio in the literature. We will then examine how methanogenesis is currently modeled in land surface models and state-of-the-art process-based methane models, and discuss efforts to bridge the divide between laboratory-scale and global-scale approaches. This will lead to a clear recommendation of a model structure for a methanogenesis module inside a land-surface model that can predict, process-based, the $CO_2$:$CH_4$ production ratio.

## 2   The complexity of methanogenesis

One of the most challenging aspects of studying and modeling $CH_4$ production in soils is its high degree of complexity, encompassing various different processes, which are, in turn, affected by a multitude of environmental factors ((Xu et al., 2015, 2016; van Bodegom and Scholten, 2001; Grant, 1998; Song et al., 2020; Sulman et al., 2022)). Methanogenesis is not one simple straight-forward process but rather an entanglement of various interacting microbial processes in the soil (Xu et al., 2015). The two main methanogenesis pathways are hydrogenotrophic and acetoclastic methanogenesis, during which hydrogen or acetate are being used as substrate by the microbes respectively, and methane is produced (Conrad, 1999). A review performed by Xu et al. (2016) found that out of the 40 investigated models only 3 represented these two major pathways. This is significant because the two processes yield different products: acetoclastic methanogenesis results in $CO_2$ and $CH_4$ production, while hydrogenotrophic methanogenesis only produces $CH_4$ (Conrad, 1999). Furthermore, the contribution from each process to total methanogenesis varies strongly between different environments (Conrad, 1999), soil depth (Liu et al., 2017), and active layer vs. permafrost layers (Song et al., 2021), among others. Considering this, the need to distinctly represent these processes in models becomes evident if a realistic portrayal of the $CO_2$:$CH_4$ production ratio wants to be achieved.

The most important environmental factors that influence methanogenesis are temperature (Yvon-Durocher et al., 2014), soil pH (Sulman et al., 2022), water table depth (Chen et al., 2021), and soil biogeochemical conditions (Philben et al., 2020). Especially temperature has a profound effect on not only microbial decomposition processes in general (Kirschbaum, 1995; Hobbie, 1996), but also on the $CO_2$:$CH_4$ ratio in particular (Yvon-Durocher et al., 2014; Roy Chowdhury et al., 2015). This is due to the different temperature sensitivities of the processes involved (Yvon-Durocher et al., 2014), though generally both $CH_4$ and $CO_2$ production experience an increase with rising temperature (Treat et al., 2015; Schädel et al., 2016). Yvon-Durocher et al. (2014) showed in their meta-analysis that methanogenesis as a whole exhibits a higher average temperature dependence than general respiration (0.98 eV vs. 0.65 eV; measured as activation energy). In fact, such differences in temperature dependence persist even down to the finest scale, with temperature determining enzyme kinetics and thermodynamics of the individual methanogenesis sub-processes (Conrad, 2023). Temperature-induced microbial community changes may lead to changes in the dominant methanogenesis pathway, moving from acetoclastic to hydrogenotrophic with increasing temperatures, thus affecting the $CO_2$:$CH_4$ ratio (Conrad, 2023).

Naturally, this level of complexity can hardly be represented in global models. In methane modeling, there exist two common ways of representing the effect of temperature (Chandel et al., 2023): the $Q_{10}$ value and the Arrhenius-type functions (Chandel et al., 2023).

$$f(T) = Q_{10}^{\frac{(T - T_{ref})}{10}} \tag{1}$$

$$f(T) = exp(\frac{\Delta E}{R}[\frac{1}{T_0} - \frac{1}{T}]) \tag{2}$$

The $Q_{10}$ parameter expresses the factor by which the reaction rate increases upon a 10 °C change in temperature (Reichstein and Beer, 2008) and it is ubiquitously used to express temperature dependency across models (Xu et al., 2016). Although, for microbial models in particular, Chandel et al. (2023) found the Arrhenius functions to be more common. Despite its widespread use, however, the $Q_{10}$ concept is very simple (Reichstein and Beer, 2008) and not without criticism, owing in parts to the large span of reported values (Wu et al., 2021). Most models put the value for methanogenesis in the range of 1.5-4 (Xu et al., 2016) – often a central value of around 2 is chosen (Riley et al., 2011; Tang et al., 2010) – which lies in the range of values reported from many lab experiments (Roy Chowdhury et al., 2015; Treat et al., 2015; Inglett et al., 2012; Su et al., 2024; Lupascu et al., 2012). Despite that, a meta analysis by Hamdi et al. (2013) showed that the entire spectrum of reported $Q_{10}$ values from lab and field studies has a large range from <2 to >300 (Hamdi et al., 2013). Wu et al. (2021) further criticized the use of constant $Q_{10}$ parameters in models as overly simplistic, even finding that the decomposition rate behaved linearly rather than exponentially in the 5 °C to 30 °C range in their model experiment (Wu et al., 2021). They argue instead in favor of a more in-depth biogeochemical model approach that accounts for individual processes (Wu et al., 2021).

As for the second frequently used method, the idea behind Arrhenius functions is to express the temperature sensitivity through the activation energy of the process in question (Yvon-Durocher et al., 2014; Chen et al., 2021; Chandel et al., 2023;

Li et al., 2023). This approach is based on fitting data to the Boltzmann-Arrhenius function, which, similar to the $Q_{10}$ approach, assumes an exponential increase of the metabolic rate with increasing temperature (Yvon-Durocher et al., 2014; Chen et al., 2021). Here, reported values for methanogenesis lie between 0.62 and 0.98 eV (Yvon-Durocher et al., 2014; Chen et al., 2021; Li et al., 2023). Both $Q_{10}$ and activation energy values have been observed to decrease with increasing temperature and vice versa (Hamdi et al., 2013; Reichstein and Beer, 2008). In models, the $Q_{10}$ parameter is usually chosen, with different processes sometimes having their own distinct $Q_{10}$ values (Song et al., 2020). This is still rare, however, with many models settling on a single $Q_{10}$ value for methane production (Riley et al., 2011; Kettunen, 2003; Xu et al., 2015), despite the evidence for differences in the temperature response between the main pathways (Conrad, 2023). Methanotrophy usually has its own $Q_{10}$ value in models, which is typically assessed at a slightly lower value than the one for methanogenesis, lying between 1.2-2.4 (Riley et al., 2011; Kettunen, 2003; Zhu et al., 2014; Sabrekov et al., 2016; Murguia-Flores et al., 2018; Grant, 1999). Since temperature is only a piece of the puzzle, the difficulty of how to accurately represent this factor in models alone hints at the overarching complexity of methane modeling.

Besides the two main methanogenesis pathways introduced earlier, there exist further processes that have an effect on methanogenesis. This can either be directly through processes like hydrolysis and fermentation, which break down the organic matter and provide the substrate for methanogenesis (Tang et al., 2016b; Grant, 1998), or indirectly through other redox reactions such as Fe(III) reduction (Sulman et al., 2022; Zheng et al., 2019; Philben et al., 2020; Yang et al., 2016; Roy Chowdhury et al., 2015). Especially the interplay between methanogenesis and Fe(III) reduction has been the subject of recent studies and their interactions have started to be included in models (Sulman et al., 2022; Zheng et al., 2019). Additionally, some soil processes are in competition with methanogenesis for substrate, like other, energetically more favorable metabolic pathways (Lovley, 1991). Another example is homoacetogenesis, through which acetate is being produced by the consumption of $H_2$ and $CO_2$. While acetate is the main substrate for methane production by acetoclastic methanogenesis, homoacetogenesis thereby reduces the substrates for hydrogenotrophic methanogens (LeeWays et al., 2022; Diekert and Wohlfarth, 1994). Looking at this web of interconnected process (Xu et al., 2015; Song et al., 2020; Sulman et al., 2022), it becomes evident that by assuming a prescribed $CO_2$:$CH_4$ production ratio in process-based models, the reliability of future methane emission projections from warming Arctic soils and thawing permafrost is highly limited.

## 3   Representation of methanogenesis in LSMs

Despite recent efforts to integrate process-based methane production in LSMs (Song et al., 2020), their representation of $CH_4$ production largely remains overly simplified (Chandel et al., 2023). This is also true for the land surface schemes that are a part of widely used ESMs, such as the ones partaking in the CMIP6 (Coupled Model Intercomparison Project Phase 6), though simulating the $CH_4$ feedback was not part of this project (Eyring et al., 2016). These models were featured in the latest IPCC AR6 report (Canadell et al., 2021), so it would be desirable if they were able to simulate methane production from thawing permafrost landscapes in a more realistic fashion that reflects the seasonality and variability observed in studies (Galera et al., 2023; Knoblauch et al., 2018; Li et al., 2023; Chen et al., 2021). This dire need to more

accurately portray permafrost carbon processes in ESMs has recently been reaffirmed by Schädel et al. (2024) who concluded that methane emissions are only represented to an "intermediate" degree in ESMs. Tightly connected aspects such as wetland distribution remain "poorly" represented (Schädel et al., 2024). The latter hints at a larger problem in regards to accurately modeling methanogenesis in soils. Methanogenesis occurs when soils become waterlogged and oxygen is eventually depleted (van Huissteden, 2021). Predicting this in models, however, has been a persistent challenge (de Vrese et al., 2021; Schädel et al., 2024). In models, this limitation of methanogenesis to anoxic conditions is usually realized through two different methods: (1)simulating the water-table in a given area and (2) explicitly modeling and tracking the $O_2$ concentration in the soil layers (Morel et al., 2019). The former case is frequently realized via a TOPMODEL approach (Beven and Kirkby, 1979), which determines the inundated areas in a grid cell (Kleinen et al., 2020), thus representing horizontal heterogeneity while the latter method represents vertical heterogeneity. Although many models settle for one of the two methods, they are not mutually exclusive. Regardless of the chosen method(s), the problem remains that soil hydrology is subject to a high degree of sub-grid heterogeneity, especially in Arctic permafrost-affected regions (Beer, 2016; Schuur et al., 2008).

In JSBACH, the land component of MPI-ESM (Max Planck Institute for Meteorology Earth System Model) (Mauritsen et al., 2019) featured in CMIP6 (Zechlau et al., 2022), methane production has been modeled through a temperature dependent partition factor which prescribes the fraction of carbon released as methane from total anaerobic decomposition (Kleinen et al., 2021) - an approach based on the CLM(4Me) model by Riley et al. (2011). The temperature dependence in their model is realized through a $Q_{10}$ factor (Kleinen et al., 2020), which leads to an increased share of methane under warming conditions. The model uses the TOPMODEL approach to calculate the inundated fraction in the grid cells (Kleinen et al., 2020).

Another example is the UK Earth System Model's LSM JULES (Sellar et al., 2019), which calculates methane production from substrate availability, temperature, and the wetland fraction of the gridbox (Clark et al., 2011; Chadburn et al., 2020) through a multilayered scheme (Comyn-Platt et al., 2018; Burke et al., 2017), using a tuned methane production scaling factor (Chadburn et al., 2020). The temperature sensitivity is modeled through an Arrhenius function and inundated areas are represented through the saturated grid cell fraction via TOPMODEL (Chadburn et al., 2020; Comyn-Platt et al., 2018). Furthermore, Chadburn et al. (2020) showed an altered version of JULES called JULES-microbe, which features a much more detailed decomposition process including hydrolysis, methanogenic microbial biomass, and microbial activity, though they do not explicitly model the two main methanogenesis pathways either (Chadburn et al., 2020). Instead they partition the produced gases equally into $CH_4$ and $CO_2$, based on the theoretically assumed 1:1 production ratio of acetoclastic methanogenesis (Conrad, 1999; Chadburn et al., 2020). Recently, the UKESM has further received an emission-driven fully coupled methane cycle (Folberth et al., 2022), showing the ongoing research development towards more in-depth methane representation.

The ORCHIDEE model is another commonly used LSM, which over the years has been updated to represent permafrost processes and high-latitude peatlands in ORCHIDEE-PEAT(Guimberteau et al., 2018; Qiu et al., 2019). It has recently received an updated methane module named ORCHIDEE-PCH$_4$, based on the scheme described by Khvorostyanov et al. (2008a, b), which uses the same temperature and soil moisture dependent function for methanogenesis as for aerobic respiration, albeit with a 10-times lower rate (Salmon et al., 2022; Khvorostyanov et al., 2008a). Temperature dependence was modeled through

**Table 1.** Overview of the main models discussed in this paper.

| Models Overview | | | |
|---|---|---|---|
| Model | Methanogenesis | Temperature | Reference |
| JSBACH3.2 | pre-set fraction, following (Riley et al., 2011) | $Q_{10}$ | (Mauritsen et al., 2019; Kleinen et al., 2020) |
| JULES | scaling factor, pre-set fraction | Arrhenius | (Sellar et al., 2019; Chadburn et al., 2020; Clark et al., 2011) |
| JULES-microbe | methanogenic microbial biomass and activity, $CO_2$:$CH_4$ partition pre-set 1:1 | Arrhenius | (Sellar et al., 2019; Chadburn et al., 2020; Clark et al., 2011) |
| ORCHIDEE-PEAT | reduced rate parameter with respect to aerobic respiration, following (Khvorostyanov et al., 2008a, b) | $Q_{10}$ | (Guimberteau et al., 2018; Salmon et al., 2022; Qiu et al., 2019) |
| ELM | pre-set fraction, following (Riley et al., 2011) | $Q_{10}$ | (Ricciuto et al., 2021; Chinta et al., 2024) |
| ELM-SPRUCE | acetoclastic and hydrogenotrophic pathways, following (Xu et al., 2015) | $Q_{10}$ | (Ricciuto et al., 2021) |
| Song et al. model for IBIS | acetoclastic and hydrogenotrophic pathways, fermentation, homoacetogenesis | $Q_{10}$ | (Song et al., 2020) |
| Sulman et al. model for PFLOTRAN | acetoclastic and hydrogenotrophic pathways, fermentation, Fe(III) reduction | CLM-CN T response function | (Sulman et al., 2022; Tang et al., 2016a) |
| Tang et al. model for CLM-CN | acetoclastic and hydrogenotrophic pathways, fermentation | CLM-CN T response function | (Tang et al., 2016b; Thornton and Rosenbloom, 2005) |
| Zheng et al. model | acetoclastic and hydrogenotrophic pathways, fermentation, Fe(III) reduction | CLM-CN T response function | (Zheng et al., 2019; Thornton and Rosenbloom, 2005) |
| Morel et al. model for ISBA LSM | reduced rate parameter, based on (Khvorostyanov et al., 2008b) | $Q_{10}$ | (Morel et al., 2019) |
| Ma et al. model for TECO | ecosystem-specific $CH_4$-release ratio parameter | $Q_{10}$ | (Ma et al., 2017) |

a $Q_{10}$ function, although the relationship is assumed to be linear instead of exponential at values below 0 °C , reaching zero at

190   -1 °C (Qiu et al., 2019; Koven et al., 2011). While the base ORCHIDEE-PEAT uses the TOPMODEL approach to determine

inundated grid cell fractions (Qiu et al., 2019), ORCHIDEE-PCH4 explicitly uses the oxygen concentration in the soil for methanogenesis (Salmon et al., 2022). The latter model has only been evaluated with data from peatlands (Salmon et al., 2022). Peatlands, however, are a very specific environment with unique features and model requirements (Mozafari et al., 2023). This limits the model's application to these areas even though Arctic methane emissions from permafrost thaw will arise from other sources as well, such as thermokarst lakes or simply from thaw and inundation of non-peatland soils (Saunois et al., 2020).

A land-surface model that has seen some recent progress in improving its methane representation is the Energy Exascale Earth System Model's (E3SM) land model (ELM) (Ricciuto et al., 2021). Originally, its methane module was based on the CLM(4Me) (Riley et al., 2011), same as for JSBACH (Kleinen et al., 2021). Since then, there have been attempts to update the methane module and include a more process-based representation of many methane processes, for example in the ELM-SPRUCE version with acetoclastic and hydrogenotrophic methanogenesis, based in large parts on the process-based methane model developed by Xu et al. (2015) (Ricciuto et al., 2021; Xu et al., 2015). This updated module, however, has yet to be incorporated into the ELM for global simulations as part of E3SM (Ricciuto et al., 2021). The current version still uses the $CO_2$:$CH_4$ ratio partition factor as proposed by Riley et al. (2011), as well as a $Q_{10}$ parameter for modeling the temperature dependence of methane production (Chinta et al., 2024). As for inundation, ELM uses a typical hydrological sub-model to calculate the spacial distribution of wetlands, however, it has recently received an updated version with a focus on wetlands called ELM-Wet, which introduces a distinct sub-grid wet-landunit that enables a more mechanistic portrayal of wetland processes (Yazbeck et al., 2025).

## 4    State of process-based models of methanogenesis at local scale applications

In contrast to global LSMs, there exist smaller process-based methane models on the lab and site scale that represent many of the processes related to methane production in much more detail (Xu et al., 2016, 2015; Grant, 1998; van Bodegom and Scholten, 2001). The process-based methane models discussed in this section include both standalone models and methane-focused modules developed for larger models, such as LSMs. In contrast to the previously discussed LSMs, which are being used in global simulations, often as part of ESMs, the models in this section were developed for site-level or lab-scale applications, with an explicit focus on methane processes. Indeed, there has been an ongoing effort to refine the modeling of methane over the decades and a plethora of models with varying complexity have emerged, with models using process-based methanogenesis representation at the top (Xu et al., 2016). It is these process-based approaches that are needed to better understand the processes underlying methane dynamics in the soil, which will then enable more accurate predictions on how these processes and, by extension, the methane budget at large will react to future climate change (Chandel et al., 2023). It should be noted, however, that many of the past in-depth methane models have been designed for environments other than permafrost landscapes, with much of the research being focused on (rice) paddy soils (Fumoto et al., 2008; van Bodegom and Scholten, 2001) and general wetland areas (Tang et al., 2010; Chadburn et al., 2020; Forbrich et al., 2024).Although process-based models should ideally be applicable across different environments, permafrost-affected soils exhibit unique properties and microbial

structures (Miner et al., 2022; Beer et al., 2022; Song et al., 2021) that are only comparable to the aforementioned ecosystems to a limited degree. Masyagina and Menyailo (2020) have shown that the methane emission patterns of permafrost-affected areas differed significantly to those of non-permafrost areas, highlighting this issue. Nevertheless, since the thorough synthesis conducted by Xu et al. (2016), this development has only continued further and in recent years some highly sophisticated methane models have been published. One such state-of-the-art model is the methane module developed by Song et al. (2020) for the IBIS terrestrial ecosystem model (Song et al., 2020). It is based on microbial functional groups, encompassing acetoclastic and hydrogenotrophic methanogenesis, fermentation, homoacetogenesis, and methane oxidation (Song et al., 2020). Mathematically, these processes are largely realized through formulas based on Michaelis-Menten kinetics (Song et al., 2020), while most of the parameter values stem from Grant (1998) and Kettunen (2003). In the decomposition cascade, the model starts with dissolved organic carbon (DOC), which is calculated from the total soil organic carbon pool (SOC) via a temperature and moisture dependent DOC:SOC ratio factor (Song et al., 2020). Acetate, $CO_2$ and $H_2$ are then produced through fermentation (Song et al., 2020). In the next step, these fermentation products act as the substrate for the two main methanogenesis pathways (Conrad, 1999) and homoacetogenesis (Diekert and Wohlfarth, 1994; Song et al., 2020).

One process that has recently started to be included in methane models more frequently is iron reduction (Sulman et al., 2022; Zheng et al., 2019). It is an energetically more favorable metabolic pathway for microbes, during which Fe(III) is being reduced to Fe(II) under anoxic conditions (Lovley, 1991). Although these processes are in competition with each other (Lovley, 1991; Sulman et al., 2022), they have been observed to occur concurrently in soils (Roy Chowdhury et al., 2015; Sulman et al., 2022), thus hinting at a more complicated interplay (Sulman et al., 2022; Zheng et al., 2019). A recent model that includes this process is the model developed by Sulman et al. (2022). It features largely the same microbial (methane) processes as the Song et al. (2020) model, minus the homoacetogenesis, in a comparable level of detail. Their model, however, adds another level of complexity by explicitly modeling the Fe(III) reduction alongside the methane processes (Sulman et al., 2022). The methane production is modeled via Monod-type equations and the interactions with Fe(III) reduction as well as the dependence of the methanogenic pathway on pH was represented (Sulman et al., 2022). They found the inclusion of other terminal electron acceptors to be important for accurate methane predictions, since Fe(III) reduction either increased or decreased $CH_4$ production, depending on how much substrate was available to the microbes (Sulman et al., 2022). These findings complement the results from Tang et al. (2016b), who also used a process-based methane model, and found that Fe(III) reduction positively impacted methanogenesis, by means of raising the pH, when substrate was not limiting (Tang et al., 2016b). Their model is an augmented version of the CLM-CN model (Thornton and Rosenbloom, 2005), which has been expanded by incorporating additional biogeochemical process from, e.g., ecosys (Grant, 1998) and the model from Xu et al. (2015).

Similarly, Zheng et al. (2019) developed a process-based methane model that uses Monod-type equations to model methanogenesis (acetoclastic and hydrogenotrophic) and features Fe(III)-reduction and fermentation (Zheng et al., 2019). They further included a thermodynamic factor to simulate the dynamic between the different redox processes (Zheng et al., 2019). In their model, hydrolysis of polysaccharides was assumed to be the rate limiting process for methanogenesis under anaerobic conditions (Zheng et al., 2019; Yang et al., 2016), which aligns with the importance of substrate availability for the methanogenesis-

iron-reduction-system found by Sulman et al. (2022). This connection has further been supported by incubation study results that also found a correlation between iron reduction, acetate production and methanogenesis (Yang et al., 2016). Fermenters prefer organic carbon compounds with low-molecular weight and the fermentation products (e.g., acetate) are required for methanogenesis (Yang et al., 2016). Consequently, this early stage of the anaerobic decomposition appears to have significant impact on the final methane production rate (Yang et al., 2016; Zheng et al., 2019). The designation of hydrolysis as the rate-limiting step has, however, been called into question by Conrad (2023), who instead argued in favor of the final steps in the methanogenesis process as being rate limiting (Conrad, 2023).

The methane model developed by Morel et al. (2019) as a module for the ISBA LSM (Noilhan and Planton, 1989) is another interesting approach. They model methanogenesis with the same 10-times lower decomposition rate, compared to aerobic decomposition, from Khvorostyanov et al. (2008a) that is also used in the recent ORCHIDEE module (Salmon et al., 2022). Aside from the usual temperature and substrate availability dependence, their model also factors in the limitation by oxygen concentration in each respective soil layer (Morel et al., 2019). Their approach of explicitly modeling $O_2$ concentration in the soil layers and its impact on methanogenesis differs from the more common approach of determining the water table level and strictly limiting methanogenesis to layers below that level(Morel et al., 2019)–an approach that has previously been criticized (Yang et al., 2017). Their model, however, does not have a representation of the two main methanogenesis pathways (Morel et al., 2019), thus reducing its complexity.

The data-constrained process-based methane model from (Ma et al., 2017) is another example for a methane module incorporated in a terrestrial ecosystem model (TECO) (Ma et al., 2017). Even though methanogenesis itself is not modeled in as much detail as other models discussed here–they used an ecosystem-specific $CH_4$-release ratio parameter with no distinction between pathways–their warming experiment resulted in an increased $CH_4$:$CO_2$ emission ratio (Ma et al., 2017). This makes the study one of the few who put a focus on the changes of this ratio.

## 5 Going forward - bridging the divide between scales

Looking at the discussed small-scale process-based methane models and global LSMs side by side, it becomes clear that they differ profoundly with regard to how detailed methane processes, especially methanogenesis, are being represented. Bridging this gap and using the process understanding gained in smaller scale process-based models have been identified as major remaining challenges for making ESMs more reliable and grounded in reality (Zheng et al., 2019; Xu et al., 2016; Chandel et al., 2023; Ricciuto et al., 2021). This development is needed, if models want to capture the highly variable $CO_2$:$CH_4$ ratios observed in the field (Galera et al., 2023) and lab (Knoblauch et al., 2018; Heslop et al., 2019).At this point, it is important to clearly distinguish between methane production and emission ratios. The high variability of $CO_2$:$CH_4$ emission ratios measured in the field is the result of many different processes (Galera et al., 2023), beyond methanogenesis. The methane has to be transported to the surface and, depending on the dominant transport mechanism, may be oxidized almost completely by methanotrophs before it can reach the atmosphere (Wania et al., 2010; de Vrese et al., 2021). Additionally, $CO_2$ emissions from other processes that happen concurrently with methanogenesis at sites with anaerobic conditions, such as Fe(III) or sulfate

reduction (Dettling et al., 2006; Sulman et al., 2022), and respiration in oxic layers also affect the $CO_2$:$CH_4$ emission ratio (Galera et al., 2023). Refining the methanogenesis process alone will consequently not be sufficient for greatly reducing the uncertainty of the emission ratio between $CO_2$ and methane at the surface. However, the modeling of methanogenesis, and by extension the $CO_2$:$CH_4$ production ratio, in the soil is already a source of uncertainty. Looking at the production ratios obtained under controlled lab conditions from Knoblauch et al. (2018), who reported values between 0.2-0.8, in contrast to the fixed ratio factors used in many LSMs (see Table 1), it becomes evident that using these fixed ratios directly leads to an increase of the uncertainty of methane release, already in the initial step. To quantify this uncertainty, especially in relation to the other processes affecting the methane budget, a dynamic process-based methane models is needed, giving further agency to its development.

First efforts in this direction are being done, with one example being the inclusion of the aforementioned model by Song et al. (2020) into a terrestrial ecosystem model. Another case is the model by Ricciuto et al. (2021), which has been included in the ELM and features a process-based methanogenesis scheme (Xu et al., 2015). Their model reproduced the observed distinct seasonality of the two main methanogenesis pathways (Ricciuto et al., 2021), showing the advantages of such a detailed representation, though their model has so far only been run on a site level scale (Ricciuto et al., 2021). These models are focused on natural wetland (Song et al., 2020) and peatland emissions (Ricciuto et al., 2021) respectively, meaning that the distinct features of permafrost-affected areas (Masyagina and Menyailo, 2020) are largely not considered in their model composition and subsequent evaluation with site data (Song et al., 2020; Ricciuto et al., 2021). Still, the ELM has recently received an improved wetland scheme in ELM-Wet and there are plans to implement the already discussed in-depth methane model by Sulman et al. (2022) in the future to further improve methanogenesis representation (Yazbeck et al., 2025).

Sulman et al. (2024) have recently performed a similar inclusion of an in-depth biogeochemical model into a LSM, featuring methanogenesis and methanotrophy among others, but their model study was concerned with and evaluated against data from coastal wetlands, which are distinct in their own right with, e.g., sulfate dynamics (Sulman et al., 2024). Modeling efforts like these are direly needed for permafrost-affected soils as well (Schädel et al., 2024), since estimations of the permafrost-carbon-climate feedback remain uncertain in both their spatiotemporal extent and magnitude (Miner et al., 2022; Nitzbon et al., 2024). Indeed, the future ratio of $CO_2$:$CH_4$ emissions is one of the key open questions in that endeavor (Schuur et al., 2022). Even though the emission ratio is affected by many other processes, as discussed above, the production ratio is an important initial step. Additionally, the representation of permafrost processes in ESMs is generally still severely lacking (Miner et al., 2022; Schädel et al., 2024), with many of the models informing the most recent IPCC report still not having permafrost processes included (Canadell et al., 2021).

More complexity or realism, in regards to how certain processes are modeled, might not always be the optimal way however. Sulman et al. (2018) argued in their meta study, for example, that the ever increasing complexity and amount of processes in SOC-focused models may in fact add to the already large uncertainty of projections, due to an increase in modeling possibilities to choose from (Sulman et al., 2018). A more concrete example would be the JULES LSM, which had in the past been enhanced with a more detailed methane soil-transport and oxidation scheme (McNorton et al., 2016). This scheme was later-on abandoned due to the overall negligible improvement in terms of making the results more accurate (Comyn-Platt et al., 2018).

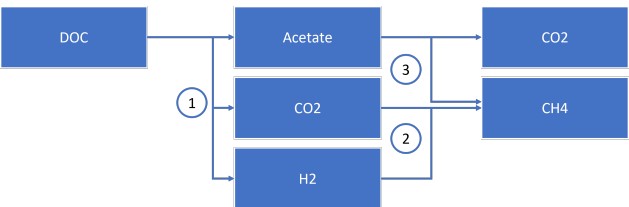

**Figure 1.** Schematic structure of the suggested core-processes required for modeling the dynamics of the $CO_2$:$CH_4$ production ratio, with (1) fermentation, (2) hydrogenotrophic methanogenesis, and (3) acetoclastic methanogenesis.

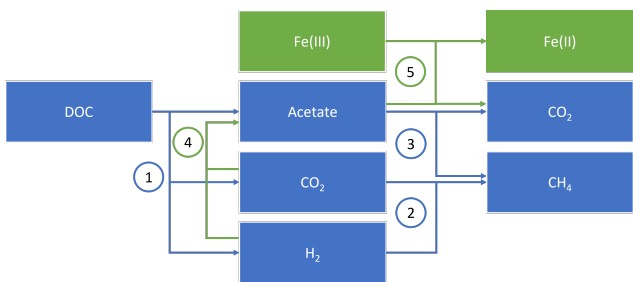

**Figure 2.** Schematic structure of a more complex approach for modeling the dynamics of the $CO_2$:$CH_4$ production ratio, with core-processes (1) fermentation, (2) hydrogenotrophic methanogenesis, (3) acetoclastic methanogenesis in blue, and closely connected process (4) Homoacetogenesis and (5) Fe(III) reduction in green.

In light of many other processes being underrepresented or all out missing in global models, the adequate complexity of each included process needs to be considered. Abrupt thaw processes, e.g., could lead to an increase in permafrost thaw emissions by up to 40% if accounted for, yet they are not featured in global models (Turetsky et al., 2020; Schädel et al., 2024). Naturally, numerical resources are not endless and current ESMs already struggle with their ever-increasing complexity (Schädel et al., 2024). Considering this, it might be necessary to find a middle ground between the current state of methane representation in most LSMs and the state-of-the-art smaller scale process-based methane models. Furthermore, it will be important to quantify the uncertainty and importance of the various processes contributing to the total methane budget, to see which processes require more attention, numerical resources and further refinement.

In conclusion, when modeling methane production in soils, the essential processes determining the $CO_2$:$CH_4$ production ratio appear to be (1) fermentation, which has been identified as a potential rate-limiting step in multiple studies (Zheng et al., 2019; Sulman et al., 2022; Philben et al., 2020), (2) acetoclastic and (3) hydrogenotrophic methanogenesis and the variable share between the two (Conrad, 1999). LSMs need to feature at least these three core-processes (see Figure 1) if the dynamics of the $CO_2$:$CH_4$ production ratio wants to be represented.

Additionally, these core-processes may be complemented by closely connected processes that either enhance or stand in competition with methanogenesis, most importantly Fe(III) reduction and homoacetogenesis (see Figure 2), something that has already been achieved in some smaller scale process-based models (Sulman et al., 2022; Zheng et al., 2019; LeeWays et al.,

2022; Diekert and Wohlfarth, 1994). Though it would undoubtedly be preferable to have these ancillary processes featured in LSMs as well, this would make the task all the more difficult. Previous studies found, for example, Fe(III) reduction to impact methanogenesis indirectly through changes to the pH (Sulman et al., 2022; Zheng et al., 2019), meaning that LSMs would have to both model global soil Fe concentrations and soil pH levels. When considering the current, highly simplified state of methanogenesis modeling in LSMs, it would be a more realistic first step to focus on the three aforementioned core-processes, before tackling further connected processes.

These processes are influenced by multiple environmental factors, the most important of which are: temperature (Yvon-Durocher et al., 2014), pH (Sulman et al., 2022), and oxygen availability (Morel et al., 2019) or water table depth (Chen et al., 2021). Soil biogeochemical conditions, especially the discussed interplay with Fe(III) reduction, is another important, albeit more complicated factor that has relatively recently emerged as a focus point in modeling studies on methane (Philben et al., 2020; Sulman et al., 2022; Zheng et al., 2019; Yang et al., 2016). Despite their importance, many of these factors are poorly quantified across the Arctic (Stimmler et al., 2023). This is largely due to the scarcity of observational field data in the vast and remote Arctic areas like Northern Russia (Suleymanov et al., 2024). ESMs, however, require spatial maps of these soil parameters to accurately portray the soil biogeochemical processes in the Arctic regions. Besides the obvious need for more field data, there are some recent publications which provide spatial datasets derived from the few data we already have. (Stimmler et al., 2023; Suleymanov et al., 2024). Stimmler et al. (2023) extrapolated sampling data to create a Pan-Arctic map of bioavailable soil elements, including Fe, based on lithology. Another interesting approach is shown in Suleymanov et al. (2024) who used machine learning algorithms to digitally map soil properties, like soil pH, in Arctic areas with scarce data availability. These techniques may prove to be important tools to bridge the large gaps in the spatial data. Both still depend on field data, however, which means that more extensive field studies remain crucial (Suleymanov et al., 2024). The same is true for methanogenesis measurement data required to benchmark models at large scales, something that is difficult to attain for the same reasons. In fact, Ma et al. (2021) have shown the importance of constraining models with in situ observational data, since $CH_4$ and $CO_2$ emissions show distinct responses to climate change (Ma et al., 2021). Even though lab incubations only offer limited insights into in situ conditions (Galera et al., 2023), they can nevertheless be useful to isolate and study single processes that are hard to disentangle in the field.

Concerning the temperature dependence, the $Q_{10}$ function is arguably the most commonly used method for describing the temperature sensitivity of methane production in models (Xu et al., 2016), likely due to its simplicity (Reichstein and Beer, 2008). At the same time, the $Q_{10}$ value has been repeatedly identified as a highly sensitive model parameter (Chinta et al., 2024; Riley et al., 2011; Song et al., 2020; Ma et al., 2017), making its accurate assessment paramount. Parameter estimations, however, vary strongly between different models (Xu et al., 2016), owing in large part to the wide range of reported values from experiments (Roy Chowdhury et al., 2015; Hamdi et al., 2013; Wu et al., 2021). Furthermore, the different temperature sensitivities of the processes involved in fermentation and methanogenesis (Conrad, 2023) need to be considered and should be represented in future models. Reducing the uncertainty introduced through the modeling of temperature dependence will be a crucial step towards improving the overall predictive abilities of methane models.

For predicting future methane emissions from soils, further processes are required. First, the transport of methane to the surface through the main three transport ways (Walter and Heimann, 2000; Wania et al., 2010; Kaiser et al., 2017) and, second, methanotrophy, which has the possibility to drastically reduce methane emissions before they reach the atmosphere (de Vrese et al., 2021). These processes are, however, already more broadly represented in models (Xu et al., 2016), including LSMs (Wania et al., 2010; Kaiser et al., 2017; Chinta et al., 2024), compared to methanogenesis. Here it could be interesting to explore, e.g., the kinetic differences between low-affinity and high-affinity methanogens, the former requiring high methane concentrations while the latter can function even under atmospheric methane concentrations (Voigt et al., 2023; Dion-Kirschner et al., 2024), which is rarely explored in models. One model study that did include high-affinity methanogens into a biogeochemical model is the one by Oh et al. (2020). They used the Terrestrial Ecosystem Model (TEM) (Zhuang et al., 2004, 2013) as a basis and found that the addition of high-affinity methanogens to the model led to a doubling of the Arctic upland methane sink, reducing net $CH_4$ emissions by ca. 5.5 Tg $CH_4$ yr$^{-1}$ (Oh et al., 2020). This significant reduction shows that further refining methanotrophy in models will also be crucial for reducing the uncertainty of $CO_2$:$CH_4$ emission ratios, and more studies focused on the inclusion of high-affinity methanogens in models are needed (Oh et al., 2020).

There are other important uncertainty sources concerning the methane budget, one of which are cold season methane fluxes, which can make up more than half of the total annual Arctic methane flux (Zona et al., 2016). In models, however, these emissions are commonly underestimated and poorly constrained (Treat et al., 2018; Ito et al., 2023). Treat et al. (2018) showed that constraining a process-based model ensemble with measured data from the non-growing season (September-May) could increase the annual wetland methane flux by 25% when compared to the unconstrained approach. These findings have been corroborated by Ito et al. (2023), who compared the cold season (September-May) methane flux outputs of 16 models to in situ observational data and found that the models underestimated methane emissions during that period, with the discrepancy being especially pronounced in months that exhibited air temperatures under 0 °C. This underestimation is due to insufficient cold season process representation and parametrization (Ito et al., 2023; Treat et al., 2018). Models fail to capture, for example, the observed burst of methane emissions during the freeze-in period in late-autumn (Mastepanov et al., 2008). This period falls into the "zero curtain" period, during which the soil stays unfrozen, while temperatures stay at around 0 °C, due to latent heat of fusion of soil water and snow cover insulation (Zona et al., 2016). The latter is especially important because changes to the snow cover affect soil thermodynamics, which, in turn, affects soil biogeochemistry and permafrost dynamics (Pongracz et al., 2021). The impact of improving the representation of snow processes in models for further reducing uncertainty in projecting Arctic methane emissions, has been shown by Pongracz et al. (2021), who implemented a multi-layer snow-scheme into the LPJ-GUESS dynamic vegetation model and found a significant improvement to the simulated permafrost extent. Further model refinement of these processes is, consequently, needed to reduce this uncertainty in the Arctic methane budget (Ito et al., 2023).

Finally, the uncertainty of wetland extent and distribution as well as their poor representation in models (Schädel et al., 2024) remain some of the most important sources of uncertainty concerning the Arctic methane budget, as recently shown again by Ying et al. (2025) in their machine-learning-based upscaling study. Here in this paper, we present a framework for a more process-based portrayal of methanogenesis in LSMs and review which processes and factors need to be considered for capturing the dynamics of the $CO_2$:$CH_4$ production ratio. This development becomes a necessity if research questions such

as the prediction of pan-Arctic greenhouse gas fluxes under a changing future hydrology want to be answered with a higher degree of confidence. However, the many other discussed processes that make up the total methane budget have high degrees of uncertainty as well and estimating their respective importance and quantifying their uncertainties will be crucial going forward.

In the end, a more process-based methanogenesis approach in models could contribute to more reliable estimates of the carbon-climate feedback, for which the relative roles of carbon dioxide and methane emissions represent an important factor (Schuur et al., 2022).

*Author contributions.* MM and CB designed the study. LK and VB contributed with ideas. MM wrote the manuscript with contributions from all co-authors.

*Competing interests.* The authors declare no competing interests.

*Acknowledgements.* We acknowledge the funding provided by the German Federal Ministry of Research, Technology and Space through the MOMENT project (03F0931A and 03F0931F).

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
