# Peer review of "Reviews and syntheses: The role of process-based modeling of the $CO_2$:$CH_4$ production ratio in predicting future terrestrial Arctic methane emissions"

_EGUsphere, 2025_

## Author Comment (AC1)

Reply to Reviewer 1 (Anonymous)
Marius Moser, Lara Kaiser, Victor Brovkin, and Christian Beer

The paper by Moser et al. is a review on the representation of methane production in process-based models. I think that in principle this is a decent overview that clearly addresses the overly simplistic way in which some models have traditionally modeled methanogenesis, while also proposing ways forward. Still, I have a number of comments for improvement and some thoughts to reflect upon.

We thank this reviewer for taking the time to carefully read our paper and to write this constructive review. The comments and thoughts provided helped to improve our manuscript. We will reply to your individual comments, with the author's response in blue underneath the respective comment.

First of all, there are many models discussed in this paper but what I'm missing is an overview table showing all LSMs and process-based methane models, and which processes are included in each. This would be really helpful to show which models are leading or still lacking. Similarly, when discussing equations, it would be good to show them. At the very least the basic  $Q_{10}$  and Arrhenius-type equations used by most models.

Thank you for the good suggestions, we will add an overview table, featuring all the discussed models, to the revised manuscript. The  $Q_{10}$  and Arrhenius equations will be explicitly shown in the text as well.

**The table will look like this:**

| Models Overview |                                            |                 |                                          |
|-----------------|--------------------------------------------|-----------------|------------------------------------------|
| Model           | Methanogenesis                             | Temperature     | Reference                                |
| JSBACH3.2       | pre-set fraction, following (Riley et al., | Q 10 | (Mauritsen et al., 2019; Kleinen et al., |
|                 | 2011)                                      |                 | 2020)                                    |
| JULES           | scaling factor, pre-set fraction           | Arrhenius       | (Sellar et al., 2019; Chadburn et al.,   |
|                 |                                            |                 | 2020; Clark et al., 2011)                |
| JULES-microbe   | methanogenic microbial biomass and         | Arrhenius       | (Sellar et al., 2019; Chadburn et al.,   |
|                 | activity, CO2:CH4 partition pre-set 1:1    |                 | 2020; Clark et al., 2011)                |

Otherwise, given the paper's title and conclusion, I don't know why the paper decided to focus only on how the models represent methane production, because this is not the only uncertainty related to modeling Arctic methane emissions. Simply knowing where wetlands are located is perhaps one of the largest uncertainties in calculating Arctic methane emissions, as most recently shown by Ying et al. (2025). Correctly simulating snow cover is also very important to accurately simulate soil temperature and soil water content, which in turn affect soil biogeochemistry (Pongracz et al. 2021). Abrupt thaw processes are only mentioned once, even though they can completely transform landscapes and therefore strongly affect methanogenesis.

I am not suggesting that the authors include a complete overview of the uncertainties for spatial upscaling, but the title claims that the CO2:CH4 production ratio is important to predict future Arctic methane emissions. Is this true when compared to accurately simulating the environmental drivers that govern methane production and consumption? I agree that we need to include the right biogeochemical processes, but if the ecological and environmental boundary conditions are misrepresented in the model then the output will be wrong regardless. I hope the authors can reflect on this, because this paper mostly gives qualitative evidence to support the importance of the CO2:CH4 ratio. Perhaps add some text that delves deeper into how this relates to other types of uncertainty in modeling Arctic methane emissions.

The points raised in these two paragraphs are very important and the authors agree that for predicting methane emissions there are many other sources of uncertainty, most notably the ones discussed in the comment. In addition to improving the representation of methanogenesis processes, other processes like methanotrophy and transport processes need to be reliably represented to predict future Arctic methane emissions. We claim that how to represent methanogenesis processes in land surface models is among the most understudied parts. Obviously, boundary conditions are also very important as the reviewer states. However, even if we had e.g., a highly accurate wetland distribution, and a reliable snow module, the pre-set methane ratio factors, used in many models, would prevent us from capturing the observed dynamic of the CO2:CH4 production ratio and hence dynamics of methane production and emission.

We agree that the other uncertainty sources were understated in the initial version of the manuscript and, following the reviewer's suggestion, we will add more text in the discussion section to better portray this. A detailed discussion of all the processes and sources of uncertainty involved in the system is beyond the scope of this paper, however.

One more nit-pick about the title is that it should be specified that this paper focuses on terrestrial methane emissions, not other methane sources in the Arctic such as lakes, geological sources, wildfires, and rivers and streams (or the ocean, for that matter). Parmentier et al. (2024) showed that these other sources can contribute over 30% of the Arctic methane budget.

Thank you for pointing this out, we will adjust the title to reflect this by adding "terrestrial" and include the cited reference in the text.:

Line 67-69: "It is also worth noting that we focus on terrestrial emissions in this study. Other methane sources, e.g., wildfires, lakes, and marine and geological sources, also make up a significant part of the Arctic methane budget, potentially contributing over 30% to it (Parmentier et al., 2024)."

In addition, there are a couple of processes that are not discussed but might be important in the Arctic methane budget. In a region where the cold season lasts for most of the year, winter emissions can become quite significant (Zona et al. 2016; Treat et al. 2018), but this is not discussed in the paper. For example, burst-like emissions upon freeze-in are a physical process that locally can lead to very high emissions (Mastepanov et al. 2008). While not directly related to methanogenesis, this is not implemented in any model and also not discussed in the paper. It shows that transport is still highly uncertain, despite what's claimed on line 66-67. See also Ito et al. (2023) for a nice overview of how the models currently represent cold season fluxes.

Thank you for discussing these processes. Cold season emissions and their representation in models are missing from this review and some text about them will be added to the revised manuscript together with the references cited.

As for the uncertainty of methane transport, we agree with this. However, the referred-to lines in the text were only meant to be seen as relative to the state of methanogenesis in LSMs which, in comparison, has arguably not seen as much development. In the introduction section of the revised version of the manuscript we will make more clear that this review is not about uncertainties in all processes governing the full methane budget but only focusing on methane production processes.

High-affinity methanogens are discussed briefly in the manuscript (line 338-341) but I think that these also warrant more discussion. The authors say that this is yet to be explored in most models but neglect to cite Oh et al. (2020) who simulated these methanotrophs and showed that the high latitude methane sink strongly increased as a result, thus reducing net emissions by ~5.5 Tg CH4 yr-1. These numbers are uncertain of course, but they stress the need to also focus on methanotrophy, not just methanogenesis. Btw, Oh et al. used a modified version of the Terrestrial Ecosystem Model (TEM), which I think is not mentioned in this paper despite a long history in modeling northern methane budgets (e.g. Zhuang et al. 2004; 2015), and one of the inspirations for the CLM(4Me) model mentioned in this paper. Might be useful to add.

As the focus of this review is on methanogenesis processes, we tried to keep the focus on the methanogenesis process in this paper so we kept this part about methanotrophs brief on purpose. However, we agree of course that methanotrophy is a very important process in its own right. We will add some further text to elaborate on this, including the reference to Oh et al. (2020), thanks.

The TEM is indeed an important model to mention and we will add this in the manuscript, thank you for pointing this out.

I know that the authors aim to focus on methanogenesis in particular, and I understand the wish for that focus, but the issues I mention above are also important to predict future Arctic methane emissions. Whether the CO2:CH4 production ratio is more important is unclear from this paper. For example, Sulman et al. (2022) did show an effect of Fe(III) reduction on these ratios, but the overall effect on emissions was relatively minor. Either a quantification of process importance or an uncertainty analysis would have been helpful to know whether including the extra complexity in the model, as suggested in Figure 2, really would lead to an important increase in model performance and reliably predict future methane emissions in the Arctic. If the authors are unable to better quantify this importance, then I suggest that the title and conclusions are adjusted accordingly.

Thank you for your comments. We indeed wanted this paper to be specifically focused on methanogenesis in order to highlight an often-overlooked aspect of methane emission modeling. Therefore, going into detail about all the other processes and sources of uncertainty in the system would be beyond the scope of this paper. But the reviewer raises an important point. As stated previously, we will include some further text in the discussion about the other processes/uncertainties that are discussed in the review to better contextualize the role of the CO2:CH4 production ratio.

As for quantifying this importance, we did not do this so far and also found little in the literature on this. However, even under controlled conditions, such as incubation experiments in the lab, the CO2:CH4 ratio varied between 0.2-0.8 (Knoblauch et al., 2018). This stands in stark contrast to the constant ratio factor, which is applied to the overall anaerobic decomposition and used in many LSMs, thus directly translating into great uncertainty of the methane production in the very first step, even before processes like transport and methanotrophy come into effect. To really estimate this uncertainty in relation to the uncertainties of other processes that govern the methane budget as a whole, we would need to apply a dynamic model that features the most important methanogenesis processes, which we discussed in the paper. We agree with the reviewer that such uncertainty analysis would be very important but for that we first need to represent the underlying processes. We will explain this need for a more process-based methane model able to predict the CO2:CH4 ratio in LSMs and the importance of quantifying the uncertainty in the revised manuscript and amend the conclusion accordingly. To reflect this adjustment, we will change the title of the paper to "The

role of process-based modeling of the CO2:CH4 production ratio in predicting future terrestrial Arctic methane emissions", as suggested by the reviewer.

**Minor comments:**

Line 42-44: are these both weight and molar ratios? Since C-CO2:C-CH4 and CO2:CH4 are both mentioned. I recommend converting these to the same unit for better comparison.

Galera et al. (2023) calculated their ratios on a molar basis while Heslop et al. (2019) calculated GHG production potentials from their incubations. Since the former measured in situ emission ratios and the latter production potential ratios, comparing these numbers is probably not that helpful and we only listed them here with the intention to show the wide spread of reported results in their respective studies.

Line 90: please show examples of a  $Q_{10}$  and Arrhenius-type equation in the text. Preferably with an example plot of how they differ.

See reply to previous comment, equations have been added to the text there based on the formulas shown in Xu et al. (2016).

$$f(T) = Q_{10}^{\frac{(T - T_{ref})}{10}}$$

$$f(T) = exp(\frac{\Delta E}{R}[\frac{1}{T_0} - \frac{1}{T}])$$

Line 144: very minor comment: maybe change "current" to "latest" (since it's been a couple of years)

Changed to "latest"

MPI-ESM (JSBACH) in particular uses methane production based on CLM(4Me) while other CMIP models (CESM2, NorESM1-ME; see table in Zechlau et al. 2022) use versions of CLM for their land modules.

The wording of this line in the manuscript is inaccurate and the sentence has been reworked to: Line 160-161: "(...) which prescribes the fraction of carbon released as methane from total anaerobic decomposition (Kleinen et al., 2021) – an approach based on the CLM(4Me) model by Riley et al. (2011).

Line 194-195: can you name these models here, and not just the references? Good to add to an overview table.

See reply to previous comment, an overview table will be added at this point in the manuscript, featuring the discussed models. The table will then be referenced here.

Line 229: which process-based model?

Tang et al. (2016) used the CLM-CN model by Thornton and Rosenbloom (2005) as a basis and extended it by incorporating processes from other models, such as biogeochemical reactions from Grant (1998) and Xu et al. (2015) and pH buffering from the WHAM model by Tipping (1994). See Tang et al. (2016) for further details.

They did not give a name to their new model version so we did not specify it in the text. We did change the line in the manuscript to:

Line 230-231: "Their model is an augmented version of the CLM-CN model (Thornton and Rosenbloom (2005) which has been expanded by incorporating additional biogeochemical process from, e.g., ecosys (Grant, 1998) and the model from Xu et al. (2015)."

Line 292-293: is this 40% of permafrost thaw emissions in the form of CO₂ or CH₄?

The number of 40% from Turetsky et al. (2020) is referring to general carbon loss (Pg C). Of these 40%, they found the share of  $CH_4$  to be ~20%, which, however, translated into 50% of the added radiative forcing (Turetsky et al., 2020).

Line 320: typo: "Arcitc"

Typo corrected, thank you.

**References**

Ito, A., Li, T., Qin, Z., Melton, J. R., Tian, H., Kleinen, T., et al. (2023). Cold-Season Methane Fluxes Simulated by GCP-CH4 Models. Geophysical Research Letters, 50(14), e2023GL103037. https://doi.org/10.1029/2023GL103037

Mastepanov, M., Sigsgaard, C., Dlugokencky, E. J., Houweling, S., Ström, L., Tamstorf, M. P., & Christensen, T. R. (2008). Large tundra methane burst during onset of freezing. Nature, 456(7222), 628–630. https://doi.org/10.1038/nature07464

Parmentier, F.-J. W., Thornton, B. F., Silyakova, A., & Christensen, T. R. (2024). Vulnerability of Arctic-Boreal methane emissions to climate change. Frontiers in Environmental Science, 12. <a href="https://doi.org/10.3389/fenvs.2024.1460155">https://doi.org/10.3389/fenvs.2024.1460155</a>

Pongracz, A., Wårlind, D., Miller, P. A., & Parmentier, F.-J. W. (2021). Model simulations of arctic biogeochemistry and permafrost extent are highly sensitive to the implemented snow scheme in LPJ-GUESS. Biogeosciences, 18(20), 5767–5787. <a href="https://doi.org/10.5194/bg-18-5767-2021">https://doi.org/10.5194/bg-18-5767-2021</a> Treat, C. C., Bloom, A. A., & Marushchak, M. E. (2018). Nongrowing season methane emissions—a significant component of annual emissions across northern ecosystems. Global Change Biology, 44, 163. <a href="https://doi.org/10.1111/gcb.14137">https://doi.org/10.1111/gcb.14137</a>

Ying, Q., Poulter, B., Watts, J. D., Arndt, K. A., Virkkala, A.-M., Bruhwiler, L., et al. (2025). WetCH4: a machine-learning-based upscaling of methane fluxes of northern wetlands during 2016–2022. Earth System Science Data, 17(6), 2507–2534. <a href="https://doi.org/10.5194/essd-17-2507-2025">https://doi.org/10.5194/essd-17-2507-2025</a>

Zona, D., Gioli, B., Commane, R., Lindaas, J., Wofsy, S. C., Miller, C. E., et al. (2016). Cold season emissions dominate the Arctic tundra methane budget. Proceedings of the National Academy of Sciences, 113(1), 40–45. <a href="https://doi.org/10.1073/pnas.1516017113">https://doi.org/10.1073/pnas.1516017113</a>

Zhuang, Q., Melillo, J. M., Kicklighter, D. W., Prinn, R. G., McGuire, A. D., Steudler, P. A., et al. (2004). Methane fluxes between terrestrial ecosystems and the atmosphere at northern high latitudes during the past century: A retrospective analysis with a process-based biogeochemistry model. Global Biogeochemical Cycles, 18(3), GB3010.

https://doi.org/10.1029/2004GB002239

Zhuang, Q., Zhu, X., He, Y., Prigent, C., Melillo, J. M., McGuire, A. D., et al. (2015). Influence of

changes in wetland inundation extent on net fluxes of carbon dioxide and methane in northern high latitudes from 1993 to 2004. Environmental Research Letters, 10(9), 095009. https://doi.org/10.1088/1748-9326/10/9/095009

---

## Author Comment (AC2)

**Reply to Reviewer 2 (Guy Schugers) Marius Moser, Lara Kaiser, Victor Brovkin, and Christian Beer**

Marius Moser et al., "Process-based modeling of the CO2:CH4 production ratio is important for predicting future Arctic methane emissions"

The manuscript by Marius Moser and coauthors provides a review of model representations of methanogenesis (CH4 production) in site-scale models and land surface schemes. It highlights the pathways of CH4 production and their difference in production of CO2 and CH4, and it highlights the importance of capturing the ratio between these two compounds for accurate assessments of climate impacts, with a focus on the Arctic.

The manuscript provides a good overview of the literature on this subject and is well-written, and the overview provided and comparison of model implementations is of interest to the modelling community. After some modifications, I would like to recommend it for publication in Biogeosciences.

However, this does not necessarily mean that I agree with the proposed strategy of refining models with this information; I see major hurdles in the scaling of information that is primarily derived at laboratory scale to models that work at field scale or even grid cells of tens of kilometers. I would like to suggest the authors to discuss this challenge in greater depth (see my comments on section 5 below). I provide further suggestions and comments, hoping that these can help to strengthen the structuring and the impact of the paper even further.

We thank you for taking the time to carefully read and review our manuscript and for providing comments that will help to improve our paper. We will reply to each respective comment underneath. The reviewer's comments are written in black font while the author's response is in blue.

**Major recommendations:**

I would recommend to bring the introduction of the different pathways (which now starts at I. 117) forward in the text. You bring up hydrogenotrophic and acetoclastic methanogenesis already in the first paragraph of section 2. Because the pathways are so fundamental for understanding your argumentation, I would suggest to start with an explanation of those in

section 2. The pathways could probably be illustrated in a simplified way, e.g. as done in Fig. 1.

Thank you for the good point, this indeed makes sense and we moved the introduction of the pathways to the first paragraph of section 2. A simplified illustration akin to Fig. 1 would probably be unnecessary though, since it would look very similar. We could refer to Fig. 1 at that point.

In section 4, the distinction between LSMs and process-based models in the paper seems somewhat arbitrary - e.g., I would group some of the models in this section also under LSMs (e.g. ISBA-LSM). Maybe it would be better to distinguish levels of complexity in different models, or application to site/point studies vs. application to regional (gridded) simulations.

It is true that the line gets blurred in a lot of cases and we think your suggestion of more clearly distinguishing levels of complexity (of the methanogenesis) and area of application is good. We will specify that the models presented in section 4 feature a process-based approach to methanogenesis in particular and that they were developed for small-scale application, in contrast to the global LSMs discussed in the previous section.

We will change the title of Section 4 to "State of process-based models of methanogenesis at local scale applications" and add this to the opening paragraph for clarification: "The process-based methane models discussed in this section include both standalone models and methane-focused modules developed for larger models, such as LSMs. In contrast to the previously discussed LSMs, which are being used in global simulations, often as part of ESMs, the models in this section were developed for local applications, with an explicit focus on methane processes."

In section 5, I think that the authors could do a better effort to bring the full complexity of the system into play. While the CO2:CH4 production ratios in methanogenesis are well represented by the two pathways that are discussed in detail in the study, the CO2:CH4 ratios measured in the field are a combination of emissions from methanogenesis as well as emissions from other processes (e.g. CO2 fluxes from heterotrophic respiration under aerobic conditions), some of which may dominate over the methanogenesis fluxes. The cited papers from Galera et al. (2023) and Schuur et al. (2022) highlight this (and Galera et al. (2023) argues in fact that incubations provide limited information on in situ conditions). It would be worthwhile to enhance the discussion on how to obtain a useful parameterization of these processes at large scales, and on the availability of relevant data for parameterizing and evaluating models (or maybe the authors could provide suggestions for relevant measurements to be undertaken to constrain such models). The discussion of CO2:CH4 ratios from methanogenesis and of CO2:CH4 ratios as measured in the field (and hence originating from multiple sources) should be disentangled more in section 5.

This distinction is indeed important and we will make this clearer in the section 5 discussion. We will also add an opening paragraph to section 5 that puts the methanogenesis processes into context with the system at large, including a clearer distinction of CO2:CH4 emission ratios and CO2:CH4 production ratios. This paper intends to focus on the methanogenesis part, i.e., the CO2:CH4 production ratio, and how it is represented in models. Discussing the entire complexity of the system is beyond the scope of this paper. The authors of course agree that the emission ratios in the field are the result of a multitude of processes, only one of which is the CO2:CH4 production ratio, and other processes like methanotrophy might locally be more influential for the final emissions. By restricting methane production to a fixed ratio factor tied to overall anaerobic decomposition, we nevertheless fail to represent the observed dynamic of methane production (Knoblauch et al., 2018) and this directly translates to uncertainty – even in this initial step.

Following the reviewer's suggestion, we will elaborate more on relevant data for parameterization and evaluation, or the lack thereof, and measurements that are needed to constrain the models.

**Some additional suggestions:**

- I. 59: The text seems to mix two impacts of vegetation on CH₄ fluxes: (1) The presence and abundance of aerenchyma affecting the transport, and (2) the provision of substrate for methanogenesis. I would recommend to disentangle these two processes a bit further in the text, because the latter is not related to transport (which is what the paragraph deals with), but with production (which is discussed in the paragraph above)

Thank you for the good suggestion, we separated these two aspects in the text and moved the part related to production to the previous paragraph on the topic.

- I. 144: Regarding the representation of  $CH_4$  production in ESMs, I think it is important to note that, in contrast to  $CO_2$  (i.e. in the C4MIP simulations), the  $CH_4$  feedback is not part of the CMIP6 simulations. But I fully support the statement that including  $CH_4$  production and its feedback to the climate system in ESMs would be desirable.

That is a good clarification to add, thank you. We changed the sentence to: line: 144: (...) such as the ones partaking in the CMIP6 (Coupled Model Intercomparison Project Phase 6), though simulating the CH4 feedback was not part of this project (Eyring et al., 2016)."

- I. 152: The two methods presented here are not mutually exclusive: the TOPMODEL approach provides a representation of horizontal heterogeneity, whereas the layering provides a representation of vertical heterogeneity. It is great to have both introduced here, but I would recommend not to present them as contrasts.

Thanks for the great point, we changed this part accordingly. We changed the line to: "The former case is frequently realized via a TOPMODEL approach (Beven et al., 1979), which determines the inundated areas in a grid cell (Kleinen et al., 2020), thus representing horizontal heterogeneity while the latter method represents vertical heterogeneity. Although many models settle for one of the two methods, they are not mutually exclusive."

It is nice to have the most commonly used models presented (I. 158 and further). It would be nice if you could refer here explicitly to the two methods introduced in I. 152, to highlight which models adopt which of the two approaches.

Thank you for the suggestion, we added this aspect to the description of the presented models.

- I. 200: The authors focus her very much on permafrost-affected landscapes, and while the different environments will certainly play a role for the parameterization of the processes, I hope that the focus on the underlying processes, which is argued for in this study, allows (in principle) an application across different environments.

Yes, that is completely true. The application across different environments should be the goal, but, as you said, if a process-based model was developed for and evaluated against data from a specific environment, like rice paddy soils, this plays a role for parameterization. The emphasis on permafrost-affected landscapes here is meant to express how these types of soils also need to be considered in these kinds of models.

It is a good idea to explicitly state this goal in the text though, we changed the text here:

line 203: "Although process-based models should ideally be applicable across different environments, permafrost-affected soils exhibit unique properties and microbial structures (Miner et al., 2022; Beer et al., 2022; Song et al., 2021) that are only comparable to the aforementioned ecosystems to a limited degree."

**Minor remarks:**

- I. 40-46: The paragraph lists a number of incubation studies with different CO₂:CH₄ ratios. How comparable are these incubation studies in their setup - would we expect similar ratios from all studies, or are the differences explained by differences in the experimental setup?

The studies listed here certainly differ in their setup – Galera et al. conducted in situ measurements of the emission ratio while Heslop et al. (2019) and Knoblauch et al. (2018) conducted incubation studies of different length – and we wouldn't expect similar ratios as a result. These studies are mentioned here with the intent to show the wide spread of ratios found in each of the respective studies, not to necessarily to compare them with each other.

- I. 50: You mention aerenchyma here without explanation - but you provide a good explanation later (I. 55). I would recommend to either remove the term here, or bring the explanation from I. 55 forward to the first time it is mentioned.

**Good point, we removed the term in line 50.**

- I. 82: The reference to Yvon-Durocher et al. (2014) is given twice in the sentence; one of the two could be removed

**True, the second reference was removed.**

- I. 179: "Naturally, the model has ..." It is not clear from the text why this is natural - I trust it is related to the study setup?

Yes indeed, it was related to the study setup and the name of the model version being ORCHIDEE-PEAT.

We dropped the "Naturally," for clarity.

- I. 190: Unclear what "proper" relates to here - "properly incorporated"?

"Proper" here was meant as it being incorporated into the ELM as part of the E3SM climate model, so that it would enable global simulations. The "proper" in the sentence was dropped for clarity and the sentence changed to:

Line 190: "(...) has yet to be incorporated into the ELM for global simulations as part of E3SM (Ricciuto et al., 2021).

- I. 229: "compliment" should read "complement"

Was changed to "complement", thank you.